# Spatial and Temporal Variations in Atmospheric Ventilation Index Coupled with Particulate Matter Concentration in South Korea

**Seoyeon Lee [1], Seung-Jae Lee [1], Jung-Hyuk Kang [1] and Eun-Suk Jang [2,*]**

[1] National Center for AgroMeteorology, Seoul 08826, Korea; sylee@ncam.kr (S.L.); sjlee@ncam.kr (S.-J.L.); jhkang@ncam.kr (J.-H.K.)

[2] National Marine Environmental Education Center, Busan 49111, Korea

[*] Correspondence: esjang@koem.or.kr; Tel.: +82-51-400-7760

**Abstract:** The spatiotemporal variations in the atmospheric ventilation index (AVI) with the particulate matter (PM) concentrations in South Korea were investigated using a regional grid model derived from the National Center for AgroMeteorology and $PM_{10}$ concentration data obtained from AirKorea and the Korea Meteorological Administration. To construct a high-resolution AVI database with 1 h time intervals and a spatial resolution of approximately 2.4 km, a medium-range prediction was performed using a regional model twice a week from December 2018 to November 2019. The resultant dataset was used to explore the seasonal patterns of the areal distribution of a novel index: Ventilation Index coupled with PM (VIP), defined by the ratio of the AVI to PM. To determine the effects of geography on the VIP, diurnal variations of the VIP were examined at three major cities in South Korea. The emphasis of the investigation was on major cities that are planned to be developed into smart cities. This study reveals the specific spatiotemporal structure of the AVI in South Korea for the first time at a high resolution and introduced the potential usefulness of the VIP. The results provide insights that could aid decision making for determining favorable locations for better air-quality cities on an annual basis and can enable the sustainable management of fine PM in and around the areas of interest.

**Keywords:** ventilation index; particulate matter; smart city; numerical model

## 1. Introduction

The Earth's atmosphere is one of many factors that sustains human life. The planetary boundary layer (PBL), the lowest layer of the atmosphere, has practical significance as a living space. Climate change and greenhouse gas emissions, which are closely linked with air pollution, are gradually becoming severe. Recent studies have found that air quality significantly affects human life [1–3]. For example, Zheng et al. [3] reported that air pollution induces negative emotions in people. In particular, on days with severe air pollution, the ratio of people engaging in impulsive behaviors increases as a result of depression and anxiety, and air pollution adversely affects cognitive activities, labor productivity, and educational performance. Thus, regions with lower are pollution and smart cities are emerging as desirable locations for human existence.

Various observations and numerical models are used to monitor air quality and meteorological factors that significantly affect human lives [4–9]. Among them, the PBL height (PBLH) plays a key role in ground air pollution because of the inversion layer directly above the PBL [6]. While in contact with the surface of the Earth, the PBL is continuously affected by friction and heat. If vigorous convection occurs, the PBLH increases, whereas if there is little convection and the atmosphere is stable, the PBLH decreases. Lee et al. [7] examined the horizontal distribution of the PBLH and the related spatiotemporal structures in the Korean Peninsula and surrounding seas in order to evaluate. The $CO_2$ dynamics

and the dispersion of air pollutant for developing a model boundary layer scheme. The PBL wind speed (PBLW) is often considered with the PBLH when investigating air quality as it determines the horizontal diffusion of pollutants, whereas the PBLH is related to the vertical diffusion of pollutants [6].

Combining the PBLW with the PBLH allows us to investigate boundary layer changes associated with topography and simply determine whether the atmosphere has the capacity to effectively disperse pollutants [9]. The atmospheric ventilation index (AVI) is the product of the average wind velocity in the mixed layer (ML) and the height of the ML considering that the smoke immediately vertically mixes within the ML [10]. The AVI has been used in multiple studies. One study that investigated the management of smoke in the event of a fire, identifying the scale and duration for which ventilation windows should be maintained [11]. Belosi et al. [12] used box models with the PBLH and PBLW for a large-scale evaluation of average pollutant and bioaerosol concentrations. Moreover, the AVI has been used to determine the possibility of a tropical cyclone and its strength, as these may occur when the ventilation index is abnormally low [13]. Most commonly, the AVI is used as part of an early warning system in air quality studies [14].

When discussing air quality, the particulate matter (PM) concentration is a major factor that cannot be overlooked. PM concentrations in South Korean cities are higher than those in other major cities in developed countries [15]. This is because large amounts of PM are emitted per unit area owing to a high population density, advanced urbanization and industrialization. South Korea is also affected by high PM emissions because it is located in a westerly wind region [15], which sometimes results in continental anticyclones. Although the PM concentrations in South Korea have decreased over the past decade owing to PM reduction policies implemented by the government, they remain relatively high. Moreover, studies have found that high $PM_{10}$ concentrations lead to higher cerebrovascular disease mortality, which increases the overall mortality risk [16,17].

There have been limited high-resolution analyses pertaining to the PBLH and PBLW, which are closely related to air quality. This study aims to obtain the AVI in South Korea considering the depth of the atmospheric ML and the average wind velocity in the ML based on high-resolution numerical modeling data. The spatiotemporal variations in the Ventilation Index coupled with PM (VIP), which is defined by the combination of the AVI with the $PM_{10}$ concentration, was also examined for South Korea. Subsequently, suggestions for planning smart cities, which refer to a relatively small developments within large urban areas, and construction designs are proposed based on an analysis of the seasonal and temporal variations in the VIP in selected major cities in the northern, central, and southern regions of South Korea.

Section 2 introduces the data used in this study and briefly describes the methodology. Section 3 presents the high-resolution PBLH, PBLW, $PM_{10}$, AVI, and VIP results of South Korea during 1 year before the onset of COVID-19. Finally, Section 4 concludes the findings of the study.

## 2. Materials and Methods

### 2.1. Data

#### 2.1.1. LAMP WRF Data

The National Center for AgroMeteorology has been producing high-resolution numerical modeling datasets under the Land–Atmosphere Modeling Package (LAMP) since 2016 [18]. The medium-range prediction of the high-resolution LAMP is based on WRF v3.7.1 and provides data for various domains, including East Asia (d01), the Korean Peninsula (d02), South Korea (d03), and regional units of South Korea. These data are used in the agriculture, forestry, and livestock industries. The LAMP can be used as an operational model for various forecast systems, including freezing and frost injuries, drought, hail, and landslides [19–21]. The data are continuously updated and improved, and their accuracy is regularly verified [22].

For this study, hourly PBLH and 4D wind vectors were obtained to determine the AVIs in South Korea based on the LAMP v1.6 domain (d03) data at a resolution of approximately 2.4 km, excluding islands. The period of study was from 00:00 (KST) 1 December 2018 to 23:00 (KST) 30 November 2019. Thus, data for 8760 h, or 1 year, were used. The period before the onset of COVID-19 was selected to discuss the air quality in an ordinary situation, considering that COVID-19 has resulted in drastic changes in human activities, such as reduced driving, transportation, and commuting activities, all of which have collaboratively led to reduced air pollutant emissions and improved air quality. The d03 domain (Figure 1a) represents South Korea and its neighboring seas, but was limited to the inland areas to harmonize the data with the observed PM concentrations.

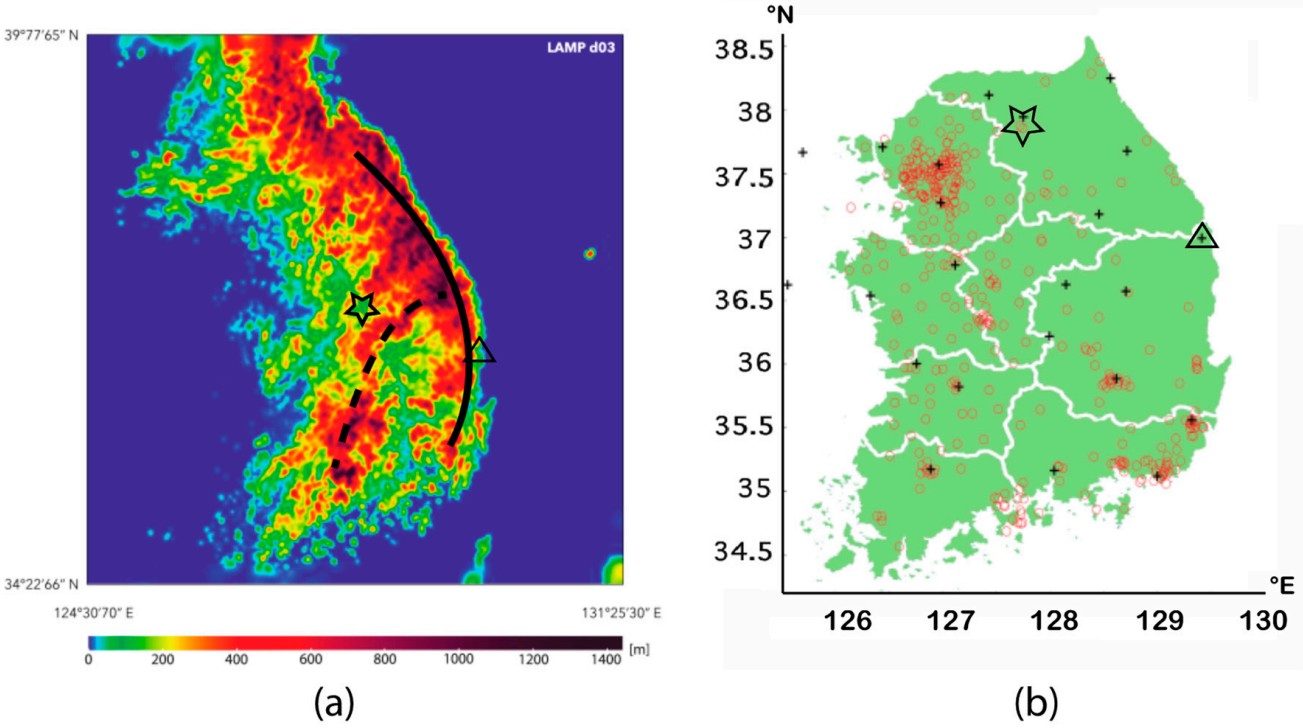

**Figure 1.** (**a**) LAMP WRF d03 domain and terrain heights, and (**b**) PM observation stations with provincial boundaries (white solid line) used in the analysis. Black solid line, Taebaek Mountains; black dashed line, Sobaek Mountains; black star, Chuncheon; black triangle, Uljin; +, KMA station; and red circle, AirKorea station.

### 2.1.2. PM Observations

The Korea Meteorological Administration (KMA) has been providing PM data along with meteorological data since 2010 [23]. However, its number of meteorological stations is limited, and they do not cover all of South Korea. Thus, the PM data of both the KMA and AirKorea were collected for use in this research [24]. The map in Figure 1b shows 427 stations, consisting of 26 KMA and 401 AirKorea stations. To examine the PM distributions of the entire Korean Peninsula, these data were statistically downscaled to the same resolution as that of the LAMP WRF using the regrid interpolation technique of the Grid Analysis and Display System (GrADS) based on the Cressman analysis scheme [25]. For observations, as with the model data, hourly data over 1 year from 00:00 (KST) 1 December 2018 to 23:00 (KST) 30 November 2019 were used.

### 2.2. Methods
### 2.2.1. AVI

The LAMP WRF model provides the calculated values of the PBLH (m), as conceptually described by Hong and Pan [26], and Lee et al. [7]. The AVI can be determined

by calculating the PBLW (m s$^{-1}$) at each grid point, considering the terrain height ($H$), as follows:

$$AVI(x,y,t) = PBLH(x,y,t) \times PBLW(x,y,t), \tag{1}$$

$$PBLW(x,y,t) = \frac{1}{N} \sum_{z=H}^{z=PBLH} \sqrt{U(x,y,z,t)^2 + V(x,y,z,t)^2}, \tag{2}$$

where $N$ is the number of vertical levels, and $U$ and $V$ represent horizontal winds.

The classification details of the AVI are listed in Table 1 [27]. A larger AVI value indicates better atmospheric dispersion throughout the PBL.

**Table 1.** Classification of the AVI, $PM_{10}$, and VIP.

| Classification | AVI | $PM_{10}$ | VIP |
|---|---|---|---|
| Very Poor | 0–235 | >151 | 0–2.8 |
| Poor | 235–2350 | 81–150 | 2.8–3.5 |
| Marginal (Normal) | 2350–4700 | 31–80 | 3.5–5.1 |
| Good | >4700 | 0–30 | >5.1 |

#### 2.2.2. VIP

This study defined a new index (VIP) by combining the conventional AVI pertaining with PM concentrations. The domestic criteria (standards in Korea) used to distinguish the $PM_{10}$ concentrations are listed in Table 1 [15]. Higher AVI values indicate greater ventilation and thereby lower $PM_{10}$ concentrations. Thus, higher AVI values indicate better air quality. Hence, the VIP is defined as follows:

$$VIP(x,y,t) = \ln\left(\frac{AVI(x,y,t)}{PM_{10}(x,y,t)} + 1\right). \tag{3}$$

A high VIP value indicates an area with a low PM concentration and good ventilation. As the classification of the VIP has the disadvantage of the presence of large intervals between different sections of the VIP, the section intervals were reduced using the natural logarithm. In addition, to prevent negative index values, the values were corrected using ln(1) = 0. VIP values ≤ 2.8 indicate poor air quality, values ≤ 3.5 indicate bad air quality, and values ≥ 5.1 denote good air quality, according to the thresholds of the AVI and $PM_{10}$ (Table 1).

### 3. Results

#### 3.1. Horizontal Distribution

Figure 2d–f show the annual average AVI, $PM_{10}$, and VIP values during the study period (December 2018–November 2019), as well as the ventilation conditions and air quality distributions over the Korean Peninsula. As the VIP is dependent on the PBLH and PBLW, indicating that it is significantly affected by the wind direction and topographical factors, it was simultaneously analyzed using these variables. The values of the PBLH is related to that of the topographical altitude, wherein it shows decreasing along mountain ranges and increasing along inland areas (between mountain ranges) (Figure 2a,c). In South Korea, there is a long mountain range (Taebaek Mountains, Figure 1, solid line) extending from the northeast to the southeast. In this mountain range, the wind velocity is low in upstream areas and high in downstream areas in the northeast (Figure 2b,c). Thus, the AVI, which is generated by combining the PBLH and PBLW, revealed high values along the mountain range, which tended to decrease as the altitude reduced toward the west owing to the greater ventilation between the mountain ranges in the central region.

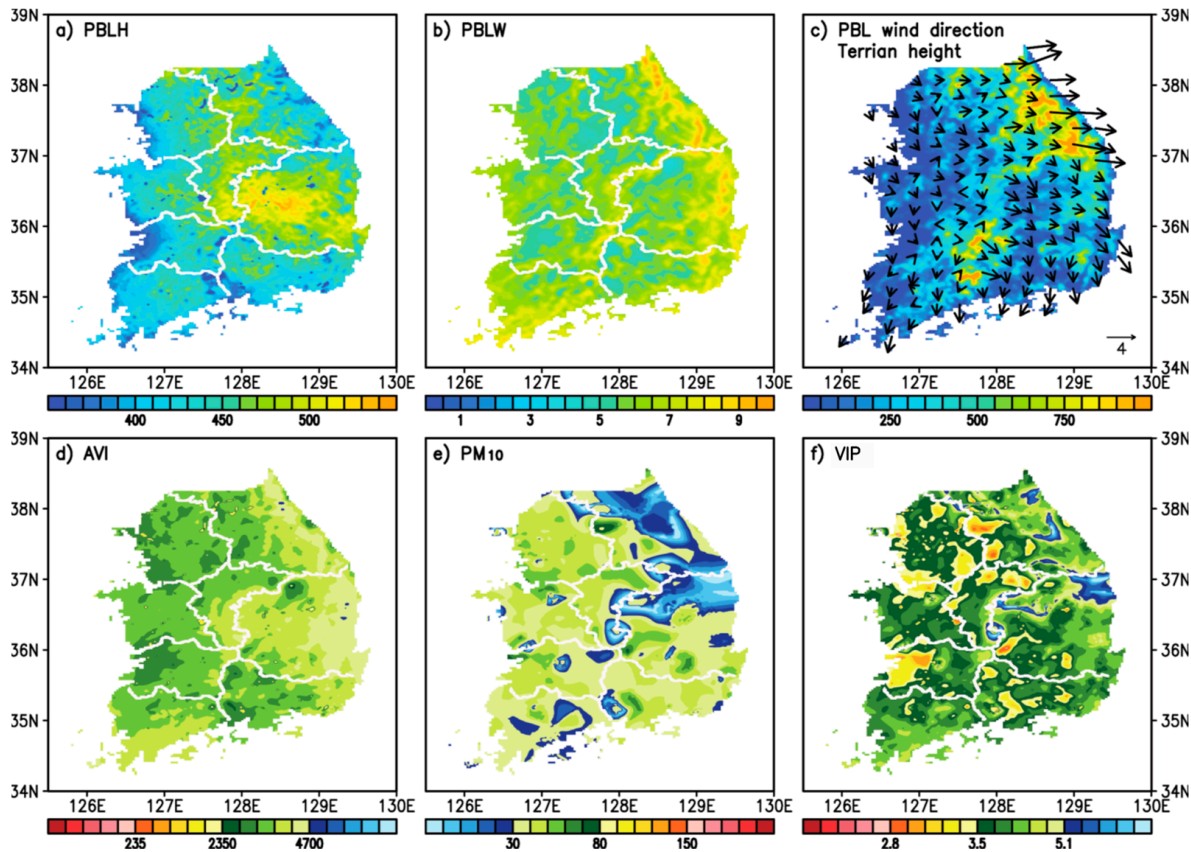

**Figure 2.** Average hourly data for one year from December 2018 to November 2019: (**a**) PBLH (m), (**b**) PBL wind speed (m s$^{-1}$), (**c**) PBL wind direction (vector) with terrain height (shaded, (m)), (**d**) AVI, (**e**) PM$_{10}$ ($\mu$g m$^{-3}$), and (**f**) VIP.

The PM$_{10}$ concentration showed a similar pattern, exhibiting low values along the mountain range. Meanwhile, the northwestern region was characterized by relatively high PM$_{10}$ concentrations. This can be explained by the inflow of pollutants from the continent via the jet stream [15] and the higher population density in these locations. The relatively high PM$_{10}$ concentration in the southeastern region can be attributed to topographical effects and northerly wind between the Taebaek Mountain Range and a the Sobaek Mountain Range (Figure 1, dashed line), which extends to the southwest from the middle section of the Taebaek Mountain Range. In addition, the annual PM$_{10}$ concentrations along the leeward side of the Taebaek and Sobaek mountain ranges were low (Figure 2c,e).

Pertaining to these patterns of the AVI and PM$_{10}$, moderate VIP values appeared throughout South Korea, except in a few areas east of the Taebaek mountain range and a few high-altitude areas. In addition, although there were regional differences, the air in the PBL exhibited smooth ventilation without stagnation, and the VIP levels gradually increase from the west to the east (Figure 2f). Meanwhile, the minimum VIP occurred near Chuncheon (127.641° E, 38.0184° N, Figure 1, black star), and the maximum occurred near Uljin (129.399° E, 37.0152° N, Figure 1, black triangle) (Table 2). This area has a basin topography surrounded by mountains, causing the PBLW to be weak, and air ventilation to be poor. Therefore, it has high PM$_{10}$ concentrations. However, caution should be maintained when building human communities on hills or mountain slopes in this area because these may result in higher traffic emissions and insufficient public transportation.

**Table 2.** Ten maximum and minimum VIP values and associated cities in the average hourly data from December 2018 to November 2019.

| | | Maximum VIP | Minimum VIP |
|---|---|---|---|
| 1 | City/County (lon, lat) | Uljin (129.399, 37.0152) | Chuncheon (126.641, 38.0184) |
| | Value | 7.066 | 2.378 |
| 2 | City/County (lon, lat) | Pyeongchang (128.725, 37.6764) | Gimje (126.791, 35.8980) |
| | Value | 7.018 | 2.524 |
| 3 | City/County (lon, lat) | Mungyeong (128.168, 36.6276) | Hwacheon (127.699, 38.1096) |
| | Value | 7.003 | 2.536 |
| 4 | City/County (lon, lat) | Yeongdong (127.992, 36.2172) | Yeoju (127.553, 37.3572) |
| | Value | 6.924 | 2.627 |
| 5 | City/County (lon, lat) | Andong (128.637, 36.6276) | Yeongcheon (128.901, 35.8752) |
| | Value | 6.839 | 2.656 |
| 6 | City/County (lon, lat) | Gunsan (126.762, 36.012) | Goyang (126.732, 37.6536) |
| | Value | 6.746 | 2.656 |
| 7 | City/County (lon, lat) | Sangju (128.051, 36.2856) | Gimhae (128.901, 35.3508) |
| | Value | 6.430 | 2.662 |
| 8 | City/County (lon, lat) | Cheorwon (127.406, 38.1552) | Hapcheon (128.227, 35.5560) |
| | Value | 6.421 | 2.712 |
| 9 | City/County (lon, lat) | Taean (126.264, 36.5592) | Paju (126.703, 37.9044) |
| | Value | 6.376 | 2.729 |
| 10 | City/County (lon, lat) | Goseong (128.549, 38.2692) | Gimpo (126.703, 37.6536) |
| | Value | 6.287 | 2.732 |

Figure 3 shows the seasonal variability of the VIP considering a three-month average, based on the distinct seasonal changes in South Korea. Seasonal patterns are significantly affected by topography, as shown by the annual averages. However, the effects vary depending on the season. In the winter, when strong northwesterlies occur owing to the effect of the Siberian air mass, the $PM_{10}$ concentrations are high in the north and northwest regions, and the PBLH is low nationwide, resulting in low VIP values mainly in the west of the Taebaek Mountain Range (Figure 3, DJF). In 2019, the $PM_{10}$ concentrations were the highest during the spring and autumn, with especially high concentrations observed in the west (Figure 3, MAM). During the summer of 2019, precipitation was low, reaching 68% of that in an average year, whereas in the autumn, precipitation was high (170% of that in an average year). Hence, the $PM_{10}$ concentrations in summer and autumn were similar (Figure 3, $PM_{10}$). Moreover, the high precipitation in the autumn influenced the PBLH, which is a function of the potential temperature, leading to low PBLH values (Figure 3, SON, PBLH). Although there were regional differences, the national average VIP was the highest in the summer, followed by that in the autumn, spring, and winter. Chuncheon, which had the lowest annual average VIP, exhibited the lowest seasonal average values,

indicating that it had low VIP values and poor air quality in all seasons except in the summer (winter 2.55, spring 3.43, and autumn 3.45).

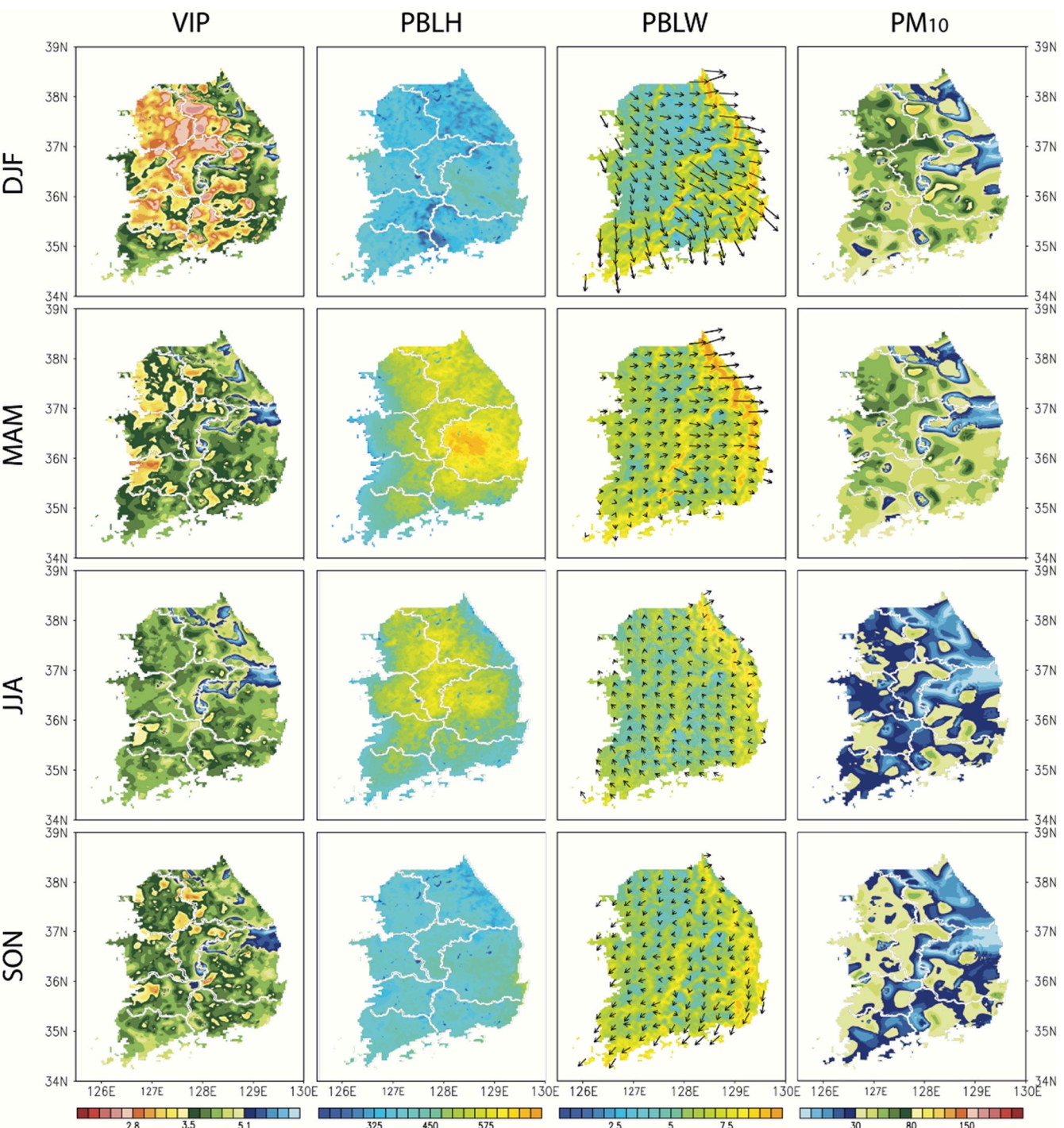

**Figure 3.** Seasonal averages for the VIP (first column), PBLH (second column, (m)), PBLW (third column, (m s$^{-1}$)), and PM$_{10}$ (fourth column, (μg m$^{-3}$)) values. First row, winter; second row, spring; third row, summer; and fourth row, autumn.

### 3.2. Spatiotemporal Analysis in Three Major Cities

As the AVI and PM$_{10}$ concentrations have significant regional differences, local spatiotemporal analysis were conducted for three cities in South Korea. Among the selected cities, Seoul, which is located in the northwestern part of South Korea, has the highest population density and is the most active region. It is characterized by mountains toward the north and south and the Han River flowing through its center (Figure 4a). Meanwhile, Sejong is an administrative city, located in a central inland area, and is preparing to be transformed into a smart city. Its residential area and central administrative agencies are located on flat lands, to the south of Sejong. There is a river flowing through this area, and a smart city is being planned nearby (Figure 4b). Finally, Busan, often referred to as the second capital, is located on the southeastern coast of South Korea and comprises a mountainous terrain. A river flows from the north to the south in the western area of Busan, and there is an industrial complex on the eastern bank of the river. Further, a smart city is being planned along the western part of Busan (Figure 4c).

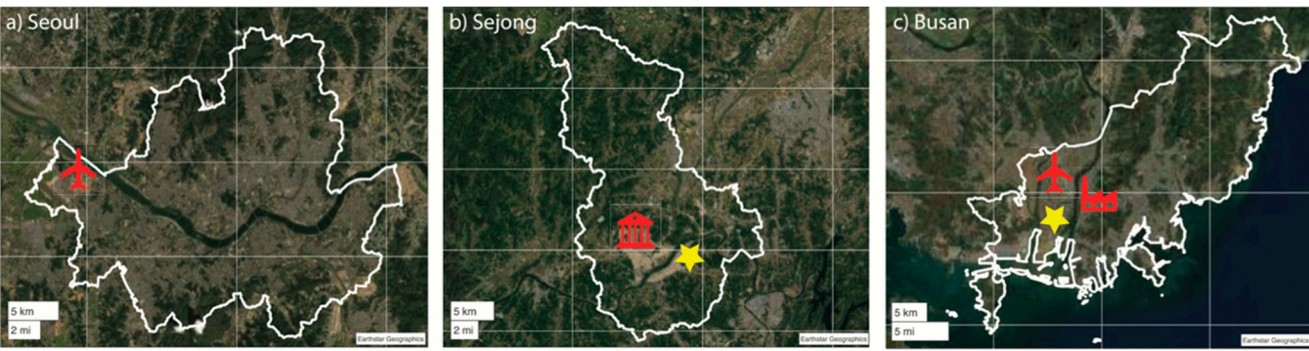

**Figure 4.** Topographical maps of three cities selected for analysis. (**a**) Seoul, (**b**) Sejong, and (**c**) Busan (from MATLAB GeoBase map). Yellow stars indicate sites where smart city transformation is planned. Red symbols indicate airports, administrative buildings, and factories.

Figure 5 shows the VIP variability in each region according to the seasonal VIP patterns of Seoul, Sejong, and Busan. It can be seen that high VIP values appear at high altitudes in mountainous areas and low VIP values occur near rivers that form valleys with low altitudes. The airport to the west of Seoul exhibits very low VIP values throughout the year (Figures 4a and 5, left). Similarly, the airport area in Busan exhibits low and moderate VIP values throughout the year, and the area with factories shows a similar pattern (Figure 5, right). In general, areas near the airport have high AVI values owing to the strong PBLW. However, the modeled PBLH is low because the effects of airport tarmac were not implemented in the model. Consequently, it exhibits a low VIP value and a high PM$_{10}$ concentration owing to airplane operation. Compared with Sejong and Busan, Seoul exhibits a more complex spatial distribution owing to its high urbanization, revealing a relatively high VIP value because of its high altitude mountainous area. The patterns of Sejong show that the VIP is dense at night in the east–west direction along the flat terrain, as shown in Figure 4 (Figure 5, middle). The maps shown in Figure 5 can help establish an urban plan or aid the construction of residential complexes in areas with low emission concentrations within these cities. Furthermore, other areas in the country may use these analysis maps as a reference for their own planning.

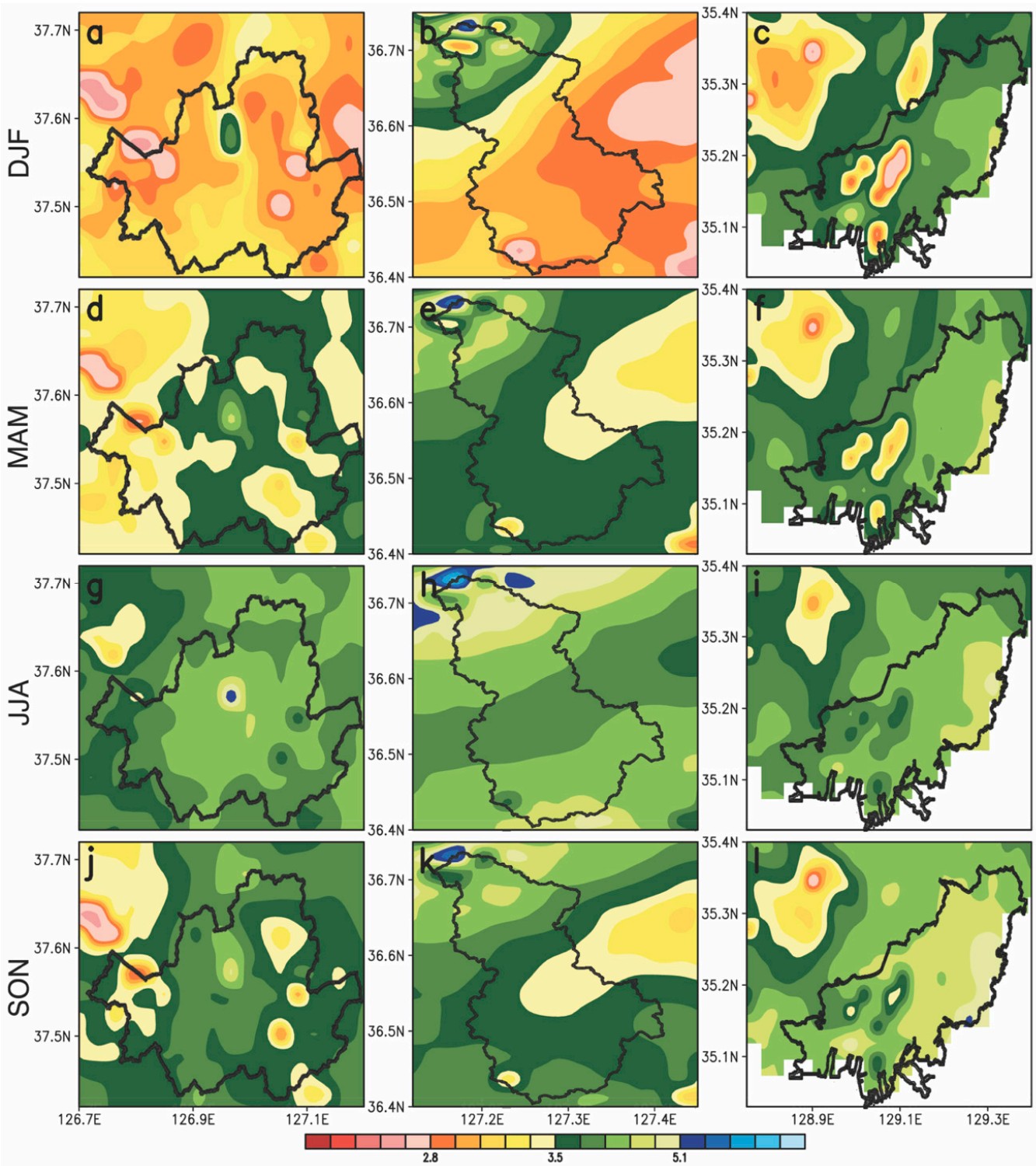

**Figure 5.** Seasonal VIPs of three major cities in South Korea. (**Left**) Seoul; (**middle**), Sejong; (**right**), Busan; first row, winter; second row, spring; third row, summer; and fourth row, autumn.

Both spatial distribution and diurnal variation are important for the VIP (Figure 6). The dashed line in Figure 6 shows that the annual average VIP values of the three studied cities were moderate (Seoul 3.73, Sejong 3.58, and Busan 4.28), among which Busan exhibited the highest value. The diurnal ranges (DRs), which are the differences between the maximum and minimum values during a day, were 2.99 for Seoul, 3.25 for Sejong, and 1.94 for Busan. As Busan showed the lowest DR, it had the smallest yearly variation. Of all seasons, winter had the lowest VIP, and summer (autumn for Busan) had the highest

DR values. Particularly, ventilation was good because the VIP values were high during 11:00–16:00 in the autumn for Busan and during 12:00–16:00 in the summer for Sejong and Seoul. The daily minimum VIP values were observed early in the day in the summer and late in the winter for each location because of the duration of sunshine. The PBLH increased at approximately 7:00, reaching a high value at approximately 14:00, while the $PM_{10}$ increased until approximately 8:00, the morning rush hour, and gradually decreases at approximately 18:00, which is the evening rush hour (figures not shown). Consistent with these findings, the VIP was the highest at approximately 13:00–15:00 and tended to decrease throughout the early morning. Compared with the other two cities, Busan is characterized by small seasonal differences, and the cause of this trend is that the PBLH is lower and diurnal variations are smaller than those of other cities. Also, it should be noted that all of the three cities show large (small) seasonal variability in the daily minimum (maximum) VIP times.

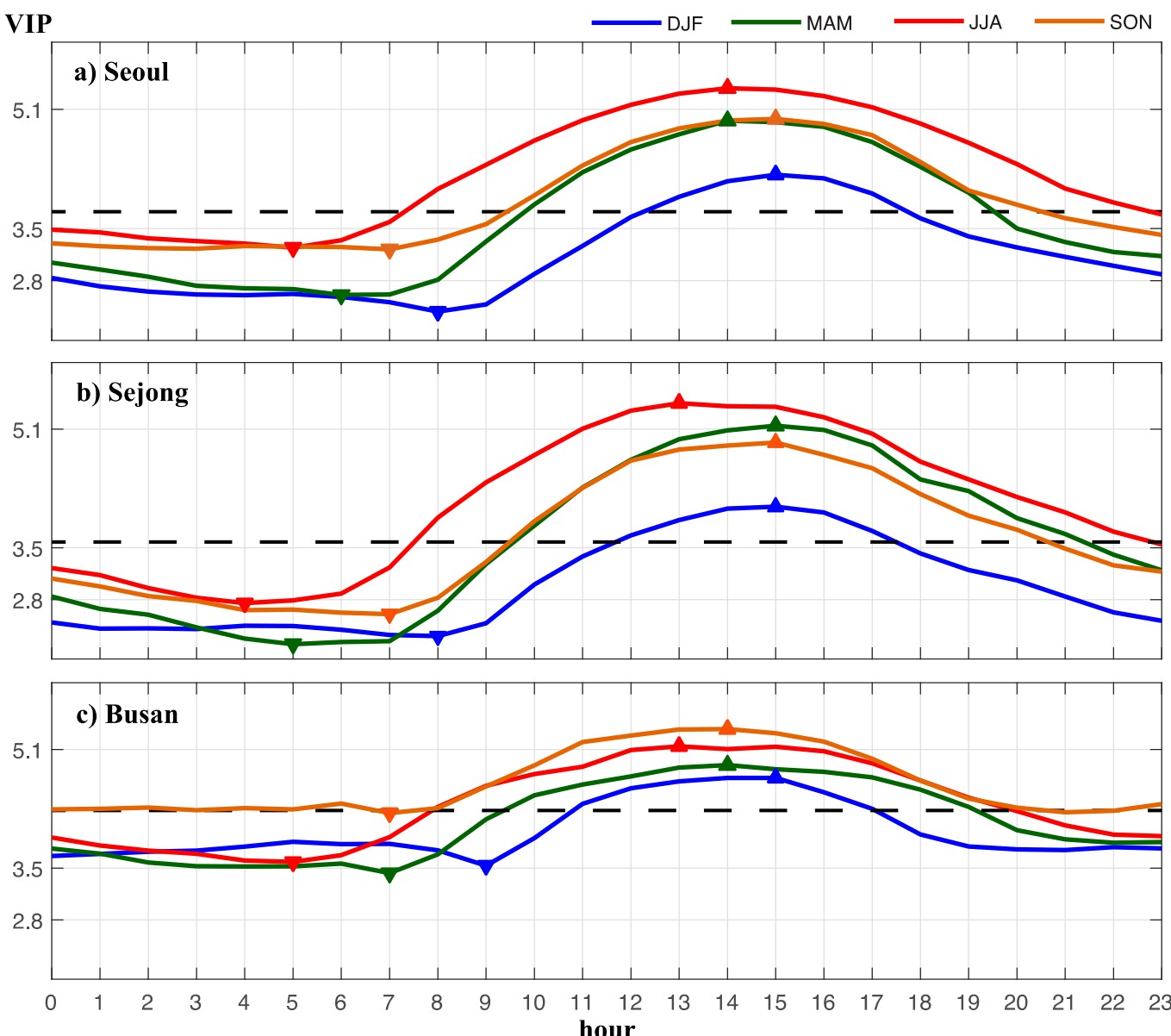

**Figure 6.** Seasonal variations in the diurnal cycle of VIP in (**a**) Seoul, (**b**) Sejong, and (**c**) Busan (blue, winter; green, spring; red, summer; brown, autumn; black dashed, annual mean; ▲, maximum; and ▼, minimum).

## 4. Summary and Concluding Remarks

This study analyzed the AVI, PBLH, PBLW, and $PM_{10}$ concentrations values of three cities in South Korea as they are closely related to the quality of life for people. In particular, regional, seasonal, and diurnal variations were examined based on the dataset of a high-resolution numerical grid model and $PM_{10}$ observations. Based on this data, a new index (VIP) was created by combining the AVI, which describes the general dispersion capacity during all stability conditions within the PBL and high-resolution $PM_{10}$ data. Using this index, the ventilation conditions and $PM_{10}$ concentrations of South Korea were spatiotemporally examined.

The VIP was significantly affected by regional and meteorological conditions owing to the geographic characteristics of South Korea (1. long mountain range in the north–south direction toward the east, 2. location on the east coast of the continent, and 3. distinct seasonal differences). As a result, the air quality was better in the summer owing to a large amount of precipitation. The VIP showed relatively high values in the spring and autumn seasons when the $PM_{10}$ concentration was sensitive to precipitation and the amount of precipitation was low. In 2019, however, owing to lower precipitation than the average in summer and greater precipitation in autumn than the average value, the VIP showed similar values in summer and autumn. Topographically, the ventilation in mountainous areas was better than that in the valleys. For a detailed comparison among the South Korean regions, spatiotemporal analyses were conducted for three representative cities in the northern, central, and southern regions. The results of this study indicate that topographic effects and the locations of airports and factories have substantial effects on the VIP. Specifically, the seasonal and diurnal variations showed that the VIP tended to decrease as the vehicle traffic increased during commuting hours [15]. In addition, the VIP was the highest at approximately 14:00, when the atmospheric inversion layer disappeared and convection was relatively active.

The VIP map can be useful for improving air quality on both national and urban scales. Further, spatiotemporal analysis of the VIP can be used to optimize conditions in other large urban areas that were not considered in this study. Furthermore, the results of this study can be used as reference data to select sites for constructing large-scale apartments in areas with low pollutant emissions. The spatiotemporal distribution of the VIP can thus help to improve urban planning in cities or provinces. However, other factors that can be altered as a result of urbanization should also be considered to evaluate their adverse influence on the quality of life. Thus, we are planning to conduct subsequent analysis on several other pollutants, including $PM_{2.5}$, $O_3$, $NO_2$, CO, and $SO_2$, in order to evaluate the costs and feasibility of converting potential sites into smart or a clean city areas. In addition, as this study used data over 1 year (2019), for a comprehensive analysis of air quality patterns; data before 2019 must also be considered. In addition, the data from 2020 should be analyzed to investigate the impact of COVID-19 and annual variations. Overall, when a smart city is planned to promote sustainable urban growth and management, the PBL climate and air quality should be examined, considering the locations of airports and factories and the topography around the target location.

**Author Contributions:** Conceptualization, S.-J.L., S.L. and E.-S.J.; methodology, S.-J.L., S.L. and E.-S.J.; software, S.L.; validation, S.-J.L. and S.L.; formal analysis, S.L. and J.-H.K.; investigation, S.-J.L., S.L. and E.-S.J.; data curation, S.L. and J.-H.K.; writing—original draft preparation, S.L.; writing—review and editing, S.-J.L. and E.-S.J.; visualization, S.L. and J.-H.K.; supervision, S.-J.L. and E.-S.J. All authors have read and agreed to the published version of the manuscript.

**Funding:** This research was funded by the "Cooperative Research Program for Agriculture Science and Technology Development (Project No. PJ014189032021 and PJ015620042021)", Rural Development Administration, Republic of Korea.

**Institutional Review Board Statement:** Not applicable.

**Informed Consent Statement:** Not applicable.

**Data Availability Statement:** The data presented in this study are available on request from the first author and [23,24].

**Acknowledgments:** We would like to thank Yun-Young So for helping with the calculation of the AVI and the three anonymous reviewers for their comments.

**Conflicts of Interest:** The authors declare no conflict of interest. The funders had no role in the design of the study; in the collection, analyses, or interpretation of data; in the writing of the manuscript; or in the decision to publish the results.

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
