# Peer review of "Spatial and Temporal Variations in Atmospheric Ventilation Index Coupled with Particulate Matter Concentration in South Korea"

_sustainability, doi:10.3390/su13168954_

Round 1
Reviewer 1 Report
The study presents a geospatial analysis of PBL, wind speed and particulate matter levels to identify areas to develop human establishments in areas favorable of atmospheric dilution to mitigate the detrimental effects of air pollution on human health. However, the conceptual framework is oversimplified and lack consideration of factors that will be altered due to the development of human establishments that can adversely influence quality of life. For example, analysis is based on PM10 only, there are many other pollutants including ozone not considered in this study. In addition, PM10 measurements are obtained in populated areas. Development of human establishments in previously native forests will always increase particulate pollution and ozone pollution. The analysis fail to recognize the costs and feasibility of such suggestions. For example, building small communities on hills or mountain slopes may result in higher traffic emissions from traffic and the inability of public transportation to serve them. The conclusions of this analysis replicate what is already known that valleys are subjected to severe air pollution episodes due to stagnation and inverse temperature layers as compared to communities in slopes and mountains. However, this type of analysis may be used to optimize conditions in large urban areas as the authors described in this study. It is recommended that authors revise the paper to present its applicability to improving air quality and quality of life in existing urban areas rather than the tool to develop new human settlements.
Reviewer 2 Report
Excellent approach, and for me, a new one. Dividing the atmospheric ventilation index by the [PM10], and resolving this new (to me) variable called VPI throughout the Korean peninsula, represents very good work. As marked throughout the manuscript, I have made a few comments and posed a question or two.

Reviewer 3 Report
General comment
The paper reports an analysis of an indicator based on coupling ventilation index with PM10 concentrations in Korea. The idea is to propose a new index linked to both aspects: pollution and ventilation. The topic could be of interest and suitable for the Journal, even if it is not very clear what this new index represent (see my specific comments). In addition, there are some other aspects not completely detailed that should be revised before considering the paper for publication.
Specific comments
Abstract (line 17), mentioning VIP without definition is not clear.
Introduction. I would suggest to mention that the ventilation index, or more in general the PBLH and the average wind speed in PBL is also used in box models for large scale evaluation of average pollution and bioaerosol concentrations as done also recently in Belosi et al. (Environmental Research 193 (2021) 110603).
Lines 109-114. The interpolation is only spatial interpolation or also temporal? What is the time resolution of the data available?
Equation (2). Why the height z start at H? What is H?
Lines 140-147. It is not very clear how these thresholds have been defined? Why exactly that numbers and not others? In addition, some discussion should be reported to explain if these thresholds could be used also in other areas or if they are site-dependent and eventually how authors suggest to define the thresholds in other conditions.
Another aspect is what the VIP represents or explains? It is mentioned that it is an indicator for “comfort”, but I believe that an high VIP could be obtained in conditions of strong winds, are we sure that strong winds are comfortable? Actually, it would be useful to explain why there is a need of such a new index and how it could be used by policy makers.
Line 248, what is DR? It is related to the max-min difference? Please explain in the paper.
Lines 269-270. I believe that this sentence should be removed. What does outdoor VIP has to do with windows? Why incineration?
Lines 285-286. I do not believe that locations of airports and factories are sensitive to VIP. Eventually, it could be exactly the opposite.
References are numbered in the text but not in the list and this make difficult to understand which reference is used in the different parts of the papers.
Round 2
Reviewer 1 Report
Not applicable
Reviewer 3 Report
Authors improved the paper during revision and answered to my questions. I believe that the paper could be accepted for publication in the present form.